# Plant sulphur metabolism is stimulated by photorespiration

Cyril Abadie[1,2] & Guillaume Tcherkez [1]*

Intense efforts have been devoted to describe the biochemical pathway of plant sulphur (S) assimilation from sulphate. However, essential information on metabolic regulation of S assimilation is still lacking, such as possible interactions between S assimilation, photosynthesis and photorespiration. In particular, does S assimilation scale with photosynthesis thus ensuring sufficient S provision for amino acids synthesis? This lack of knowledge is problematic because optimization of photosynthesis is a common target of crop breeding and furthermore, photosynthesis is stimulated by the inexorable increase in atmospheric $CO_2$. Here, we used high-resolution $^{33}$S and $^{13}$C tracing technology with NMR and LC-MS to access direct measurement of metabolic fluxes in S assimilation, when photosynthesis and photorespiration are varied via the gaseous composition of the atmosphere ($CO_2$, $O_2$). We show that S assimilation is stimulated by photorespiratory metabolism and therefore, large photosynthetic fluxes appear to be detrimental to plant cell sulphur nutrition.

[1] Research School of Biology, Australian National University, Canberra, ACT 2601, Australia. [2] Present address: IRHS (Institut de Recherche en Horticulture et Semences), UMR 1345, INRA, Agrocampus-Ouest, Université d'Angers, SFR 4207 QuaSaV, 49071 Angers, Beaucouzé, France. *email: guillaume.tcherkez@anu.edu.au

Sulfur is a crucial microelement for plant nutrition, required by the biosynthesis of sulfolipids, antioxidants, cofactors, secondary metabolites, and amino acids that are strictly or conditionally essential for human nutrition (cysteine and methionine)[1,2]. Also, grain S content is essential to form protein disulfide bridges and dictates flour processing properties (such as dough extensibility) in bread, pasta or biscuit industry[3]. Sulfur fertilization of crops represents a demand of ≈15 Mt S and an expenditure of ≈$25B each year globally[4]. It is often assumed that S assimilation positively correlates with photosynthesis, because sulfate incorporation is stimulated by sugars in plant roots[5] and inhibited under non-physiological conditions where $CO_2$ is omitted from the atmosphere ($CO_2$-free air)[6]. Nevertheless, the relationship with photosynthesis has never been tested experimentally and thus it is presently uncertain as to whether S assimilation can be impacted by environmental conditions that affect photosynthetic metabolism.

As mentioned above, solving the question of a possible correlation between photosynthesis and S metabolism in the long term is critically important because $CO_2$ mole fraction in Earth's atmosphere inexorably increases and favours plant photosynthesis, and also current metabolic engineering aims to increase photosynthesis by suppressing photorespiration[7,8]. In the past 20 years, it has been found that plants cultivated in a $CO_2$-enriched atmosphere contain less elemental S[9–13] including in grains, thereby affecting flour quality[14,15]. Accordingly, cultivation under $CO_2$-enriched conditions has been found to alter the content in S-containing antioxidant (glutathione)[16,17]. Therefore, present atmospheric changes (probable $CO_2$ doubling by 2100) seem to be detrimental to S metabolism. This problem may worsen in the near future because of limited available S in the environment, due to decreased pollution-driven $SO_2$ emissions and lower utilization of S-containing fertilisers (such as superphosphate). However, possible mechanisms explaining the lower S content when plants are grown at high $CO_2$ have not been studied.

The question of a possible correlation between photosynthesis and S assimilation in the short term is also important since in the field, photosynthesis can vary considerably depending on environmental conditions, and thus plant S nutrition could be affected accordingly. Furthermore, S metabolism is involved in electron consumption and redox metabolism in illuminated leaves and thus can affect leaf photosynthetic capacity[18]. In principle, changes in S metabolism when photosynthesis varies could stem from metabolic interactions with photorespiration. The key enzymatic activity of photorespiratory metabolism is the conversion of glycine to serine (Fig. 1). In this biochemically complicated reaction, a glycine molecule is cleaved thereby liberating $CO_2$ and ammonium $(NH_4^+)$, and a one-carbon unit is fixed onto another glycine molecule to form serine. This reaction involves tetrahydrofolate ($H_4F$) as a cofactor to transfer the one-carbon unit[19]. One-carbon ($C_1$) metabolism is thus directly involved in photorespiration (due to $H_4F$ requirement) and this is visible in, e.g., mutants affected in formyl tetrahydrofolate deformylase (which regenerates $H_4F$) that have growth defects and accumulate glycine to very high levels under photorespiratory conditions[20]. Also, $C_1$ metabolism is essential not only for photorespiration but also for many one-carbon requiring reactions of metabolism, including methionine synthesis[21]. The involvement of N-containing compounds (ammonium, glycine, serine) in photorespiration explains why photorespiration is intimately linked to enzyme activities of N assimilation and how it may stimulate nitrate reduction[22,23]. There is no such direct relationship with sulfur but presumably, photorespiration may stimulate S assimilation because serine and one-carbon units are the building blocks used to synthesize S-containing amino acids methionine and cysteine (Fig. 1). Previous experiments with glycine leaf feeding in the dark or manipulation of glutathione synthesis (via enzyme overexpression) have suggested that glycine provision by photorespiration might be of importance for glutathione synthesis and thus S metabolism[24,25]. Also, under salt stress, the increase in glutathione content has been proposed to be due to augmented photorespiration rates, which increase metabolic availability in glycine and serine[26]. Furthermore, the provision in redox power (in the form of NADH or via the malate valve, for nitrate and sulfate reduction), and aspartate (for methionine synthesis) may in principle benefit from photorespiration[27,28]. When photorespiration is low, N and S assimilation is believed to be downregulated in leaves because there is an increase in nitrate reduction in roots resulting in a generally lower N/C ratio in leaves and higher xylem transport of organic N from roots to shoots[29] and furthermore, $C_3-C_4$ Flaveria interspecies grafts suggested that in $C_4$ plants (where photorespiration is suppressed), glutathione is preferentially synthesized in roots[30]. Conversely, the response to S-deficiency includes an increase in photorespiratory transcripts and metabolites[31] and an increase in the photosynthetic $CO_2$ compensation point[32], suggesting that photorespiration participates in S homeostasis.

Still, there is presently no answer to the question of short-term metabolic interactions between photorespiration and S assimilation[33]. Here, we used a low S-demanding $C_3$ crop, sunflower, and isotopic methods to probe directly the response of the S assimilation flux from sulfate to cysteine and methionine, in vivo. Leaves from plants cultivated in S-sufficient conditions were labelled with both $^{13}CO_2$ and $^{33}S$-sulfate under controlled atmospheric $O_2$:$CO_2$ environment, including situations that favour or disfavour photorespiration compared to ambient conditions. Here, we used the $^{33}S$ isotope to allow simultaneous analysis by nuclear magnetic resonance (NMR) and mass spectrometry ($^{34}S$ is not visible using NMR and $^{35}S$ is radioactive). We took advantage of quantitative $^{33}S$-NMR to measure precisely the amount of sulfur incorporated, high-resolution liquid chromatography/mass spectrometry (LC-MS) to determine populations of isotopic species (isotopologues), and gas chromatography/mass spectrometry (GC-MS) for metabolic profiling. Because of the relatively low cysteine and methionine amount in leaves, the modest sensitivity of $^{33}S$-NMR analyses and also the fact that sulfur assimilation represents a small metabolic flux (about 0.01 µmol m$^{-2}$ s$^{-1}$, that is, between 0.05 and 0.1% of photosynthesis), analyses required a considerable acquisition time, representing a total of nearly 3000 h for isotopic measurements.

## Results

### $^{13}C$ enrichment in methionine, cysteine and their precursors.

We found that leaf cysteine inherited carbon atoms derived from photosynthesis as shown by the clear $^{13}C$-enrichment regardless of gas-exchange conditions, including at low photorespiration (inlet air at 0% $O_2$) (Fig. 2a). Methionine was much less $^{13}C$-labelled (a few %) when all C-atoms are averaged. However, LC-MS analyses with fragmentation showed that the intramolecular isotopic distribution was highly heterogeneous, with a strong $^{13}C$ enrichment in the methyl group and very little $^{13}C$ in other C-atoms (Fig. 2a, Supplementary Fig. 1). Such a pattern suggests a higher turnover of the methyl group (demethylation to homocysteine, and methionine resynthesis) and is typical of photorespiration, which leads to the production of highly $^{13}C$-enriched glycine, serine and thus one-carbon units. In fact, the isotopic enrichment in serine and the methyl group of methionine varied rather similarly (Fig. 2a–b). Also, the $^{13}C$-enrichment in the pool of metabolically active glycine and one-carbon units was always very high (≥80%; Supplementary Figure 2A-B) while the

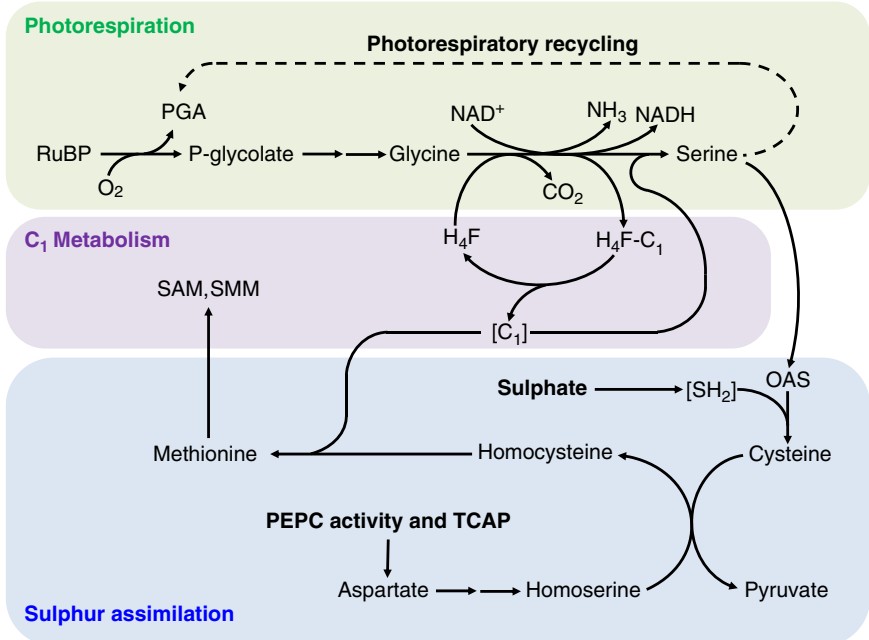

**Fig. 1** Rationale of metabolic interactions between photorespiration and S assimilation in plant leaves. This simplified scheme shows why photorespiration could be beneficial to sulfur assimilation. Photorespiration drives the production of serine that is used to synthesize cysteine de novo. In addition, photorespiration involves $C_1$ metabolism from which methyl-tetrahydrofolate ($H_4F$) can be used to synthesize methionine de novo. OAS O-acetylserine, PEPC phospho*enol*pyruvate carboxylase, PGA 3-phosphoglycerate, RuBP ribulose 1,5-bisphosphate, SAM S-adenosyl methionine, SMM S-methylmethionine, TCAP tricarboxylic acid pathway ("Krebs cycle"). For simplicity, the SAM-SMM cycle that can reform methionine is not shown

precursor of non-methyl C-atoms of methionine, aspartate, was never $^{13}$C-enriched above 18% and hardly $^{13}$C-enriched when photorespiration was suppressed under $N_2$ (0% $O_2$) as a background gas (Fig. 2b).

**$^{33}$S enrichment in methionine, cysteine and sulfate**. S reduction and assimilation appeared to be tightly coupled to the utilization of C-atoms to form cysteine, since the $^{33}$S-enrichment in cysteine strictly followed the $^{13}$C-enrichment (Fig. 2d). The $^{33}$S-enrichment in methionine was always found to be small (because of the isotopic dilution in pre-existing leaf methionine pool) and decreased at low photorespiration. Measurements of the $^{33}$S absolute content clearly showed that at high photorespiration (low $CO_2$ or high $O_2$), there was a significant increase in $^{33}$S-methionine and $^{33}$S-cysteine content (Fig. 2e). These variations were not due to changes in source sulfate, the % $^{33}$S of which remained constant (Fig. 2d). Similarly, variations in % $^{13}$C were not due to changes in the availability of source $^{13}$C since photosynthesis showed a pattern almost opposite to that in cysteine, with higher $^{13}$C assimilation under high $CO_2$:$O_2$ conditions (Fig. 2f).

**Relationship with photorespiration**. The S isotope signature (absolute $^{33}$S content and % $^{33}$S) was then used to compute the sulfur assimilation flux (i.e., sulfate reduction and synthesis of cysteine). There was a very clear linear positive relationship between the relative photorespiration rate ($v_o/v_c$) and sulfur assimilation and a negative linear relationship between sulfur assimilation and photosynthesis (Fig. 3). The $^{13}$C enrichment and isotopomer populations in glycine, serine (Supplementary Fig. 2) were used to calculate de novo cysteine synthesis and results were virtually identical to those obtained with $^{33}$S, that is, with a higher flux at high photorespiration (Supplementary Fig. 3). Consistently, the simultaneous increase in the turn-over of one-carbon units and sulfur assimilation led to a higher content in

doubly labelled methionine (that is, [$^{33}$S,$^{13}$C$_{methyl}$]-methionine) at high photorespiration. Also, the probability of finding $^{33}$S-molecules amongst the population of $^{13}$C-labelled cysteine and methionine molecules was significantly lower at low photorespiration (Supplementary Fig. 3).

## Discussion

Altogether, our data clearly show that in the short term, sulfur assimilation from sulfate and de novo cysteine and methionine synthesis respond positively to the $O_2$:$CO_2$ ratio of the environment and thus to photorespiratory activity. It is also possible that at very low $O_2$ (inlet air at 0% $O_2$), the small flux to S assimilation is explained not only by low photorespiration but also the downregulation of the aspartate pathway of amino acid synthesis (and thus methionine production) under hypoxic conditions[34,35]. A stimulating effect of photorespiration has also been found on N assimilation, with less nitrate reduction at high $CO_2$[22,23] and higher de novo glutamate synthesis as photorespiration increases[36]. In the case of nitrogen, this effect is partly driven by the increased demand in glutamate to sustain photorespiratory metabolism due to both the direct involvement of 2-oxoglutarate/glutamate cycling and the slight incompleteness of glycine-to-serine production, which must be compensated for by an increased N assimilation[37]. By contrast, sulfur metabolism is not part of the photorespiratory cycle and thus the stimulating effect of photorespiration must be due to other mechanisms. Since our experiments demonstrate an effect in the short term, it is unlikely that drastic changes in transcriptional activity and therefore in enzyme quantities (such as methionine synthase or cysteine synthase complex) are involved—although we recognize that the transcription of genes encoding adenosine 5′-phosphosulfate reductase (APR) can vary significantly within 2 h, as shown for the dark-to-light transition[38]. Rather, rapid metabolic effects must cause the stimulation of sulfur assimilation by photorespiratory conditions. High photorespiration leads to an

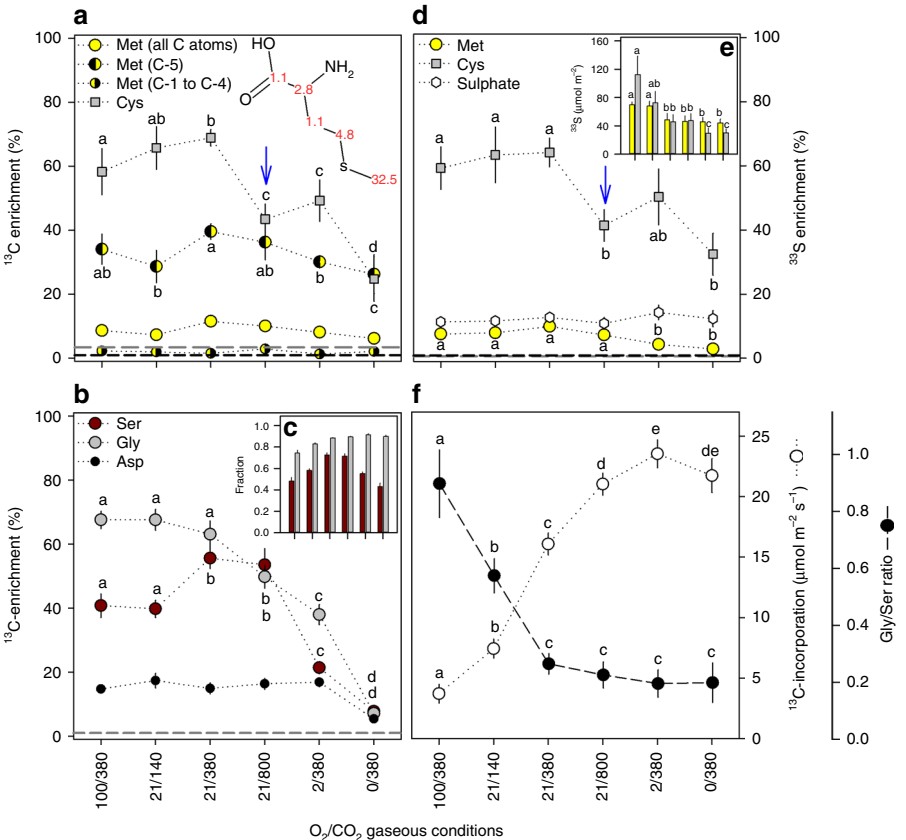

**Fig. 2 Isotopic pattern in metabolism upon double $^{33}$S-sulfate/$^{13}$CO$_2$ labelling in leaves under varying gaseous conditions (O$_2$/CO$_2$): a** $^{13}$C-enrichment measured by LC-MS in cysteine and methionine, with average positional values (across all gaseous conditions) obtained via fragmentation (LC-MS$^2$) in red; **b** $^{13}$C-enrichment in methionine and cysteine precursors (glycine, serine and aspartate); **c** mole fraction of totally $^{13}$C-labelled amongst $^{13}$C-containing molecules; **d** $^{33}$S-enrichment and **e** absolute $^{33}$S content in cysteine, methionine, and sulfate determined by LC-MS and $^{33}$S-NMR; **f** net CO$_2$ assimilation ($^{13}$CO$_2$ incorporation rate) and glycine-to-serine ratio (mol mol$^{-1}$). In **(a–d)**, the average isotopic enrichment found using natural sulfate and CO$_2$ is shown with horizontal dashed lines. Data are mean ± SE ($n = 7$ biological replicates). Note the considerable $^{13}$C-labelling in the methyl group of methionine in **(a)** while other positions are nearly indistinguishable from natural abundance. The coupling between $^{13}$C and $^{33}$S incorporation is visible in **(a)** and **(d)** as well as the general increase in sulfur incorporation as O$_2$/CO$_2$ increases **(e)**. As expected, the $^{13}$C-enrichment in precursors of methionine and cysteine C atoms is influenced by both their synthesis by photorespiration (increasing glycine-to-serine ratio) and the overall $^{13}$C input by photosynthesis **(f)**

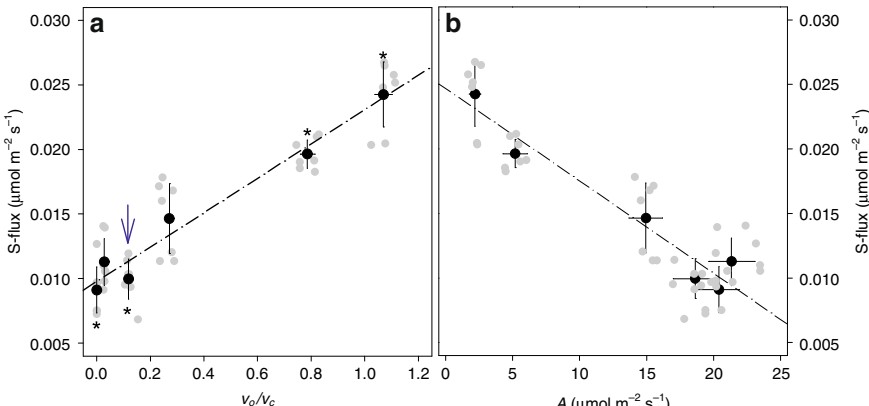

**Fig. 3 Relationship between sulfur assimilation and photosynthesis:** linear positive relationship with the relative rate of photorespiration (oxygenation-to-carboxylation ratio, $v_o/v_c$; **a** and linear negative relationship with net CO$_2$ assimilation (**b**). In (**a**), asterisks stand for statistically significant difference with 'standard' ambient conditions (21% O$_2$, 380 μmol mol$^{-1}$) CO$_2$ and the value at high CO$_2$ is labelled with a blue arrow. Sulfur assimilation declines when photosynthesis increases and photorespiration increases. Linear regressions are significant (F-test, $P < 0.01$, $R^2 = 0.95$). Data are mean ± SE ($n = 7$ biological replicates). Individual data are shown in grey

increased availability in cysteine precursor, serine. In addition, 5,10-methylene tetrahydrofolate formed by photorespiratory metabolism is more easily interconvertible to 5-methyl tetrahydrofolate because photorespiration generates NADH in the mitochondria, thereby facilitating methionine synthesis[21]. Photorespiration is also associated with antioxidant metabolism and in fact, glutathione tended to be less abundant under oxygen-depleted atmospheres (Supplementary Fig. 4). Thus, S assimilation was stimulated by both supply and demand when photorespiration increased. There is a strict requirement in photosynthetic carbon fixation for S to be assimilated, as suggested by experiments in a $CO_2$-free atmosphere[6]. Here, we found that most $^{33}$S-labelled molecules were effectively $^{13}$C-labelled (Supplementary Fig. 3). However, the metabolic commitment to S assimilation was finely regulated as photosynthesis varied (Fig. 3) and metabolic effectors certainly played a role. O-acetyl-serine (OAS), cysteine and glucose are recognized effectors under S-deficiency[39]. Here, high photorespiration was associated with a significant increase in leaf glucose-to-sucrose ratio and a decrease in OAS content (while intermediates of methionine synthesis, homoserine and cystathionine, did not change significantly) (Supplementary Fig. 4). This situation is typical of increased cysteine synthesis and limiting OAS provision, with a stimulation of OAS-thiol lyase and inhibition of S-acetyl transferase[40]. The balance between glutathione and cysteine have also been suggested to play a role in the expression of sulfate transporters[41], but here, changes in cysteine or glutathione are insignificant (cysteine) or small (glutathione) (Supplementary Fig. 4). Photorespiration also involves $H_2O_2$ as a byproduct, which has been found to stimulate both chloroplastic cysteine synthesis and sulfate reduction by APR[42].

Of particular significance is the lower $^{13}$C and $^{33}$S content in cysteine (Fig. 2a, d) and a slightly below-the-line flux of S assimilation at high $CO_2$ (arrow, Fig. 3). Accordingly, leaves were generally less rich in methionine and associated metabolites (homoserine, cystathionine) at high $CO_2$ (Supplementary Fig. 4). This depressing effect of high $CO_2$ seen here in the short term may also be relevant when plants are grown under high $CO_2$ conditions, since $CO_2$-enrichment is generally detrimental to S content in tissues[11], including in trees[17]. Also, in the long term, high $CO_2$ is associated with a decrease in expression of genes encoding proteins involved in S acquisition such as sulfate transporters, and enzymes of glutathione metabolism[43]. Of course, in the field, this effect must be modulated by soil S content as well as N and water availability. In particular, drought conditions lead to low $CO_2$ conditions (because low stomatal conductance reduces internal $CO_2$ mole fraction) and promote photorespiration. If the stimulating effect of photorespiration we see here also applies during drought conditions, it should be associated with an increased S assimilation[18]. In fact, an increase in methionine content has been found in water-stressed leaves[44] and it correlates to drought tolerance in wheat[45]. Our results also raise the question of whether S assimilation could be linked to the photosynthetic metabolic type ($C_3$, $C_4$) since it directly impacts on photorespiration activity. Can the beneficial effect of photorespiration on S assimilation contribute to explaining photorespiration persistence in the $C_3$ lineage? In fact, despite its negative impact on plant carbon balance, photorespiration has not been eliminated during plant evolution, except in species that contain a carbon-concentrating mechanism such as $C_4$ plants. While the histological location of S-assimilating enzymes in $C_4$ plants is rather controversial[46,47], recent data on transcript levels of genes encoding for APR suggest that the flux of S assimilation is actually higher in $C_4$ species compared with their $C_3$ relatives. This might be due to the higher demand in reduced S caused by

recurring oxidative stress in dry and warm habitats where $C_4$ plants grow naturally[48].

Taken as a whole, in illuminated leaves, the S assimilation flux appears to be dictated by metabolic fluxes in photosynthesis and photorespiration, which are directly determined by the gaseous composition of the atmosphere. The metabolic interaction we described here has consequences for plant physiology since drought and high temperature are common situations that may happen in the field and impact on atmospheric and/or intercellular gaseous composition ($O_2$:$CO_2$ ratio). Also, the interaction between photorespiration and sulfur metabolism opens the questions of whether S-assimilation has to be examined more closely in plants where photorespiration is manipulated by molecular engineering[7,8], or whether changes in S-fertilization practices have to be anticipated due to future Earth's atmosphere composition.

## Methods

**Plant material.** Sunflower seeds (*Helianthus annuus*, var. XRQ) were sown in potting mix and after 14 days, plantlets were transferred to 15 L pots filled with Martins potting mix (made of coir, sand, composted bark fines, and a fertiliser mixture containing gypsum, superphosphate, iron sulfate and Magrimax®) at 0.21% S (i.e., 6.3 g $SO_4^{2-}$ $kg^{-1}$). That way, plants had a sufficient available sulfate content to grow. On average, mature leaves used in this work contained 2.9 mmol $m^{-2}$ free $SO_4^{2-}$. Plants were grown in the greenhouse under 24/18 °C, 60/55% relative humidity, 16/8 h photoperiod (day/night), with natural light supplemented by Lucagrow 400 W sodium lamps (JB Lighting, Cheltenham, Australia). Plants were watered every 2 days supplemented once a week with 1.5 g $L^{-1}$ nutrient solution Peters® Professional Pot Plant Special (Everris, Netherlands) with a N/$P_2O_5$/$K_2O$ composition of 15/11/29 (and a nitrogen balance nitrate/ammonium/urea of 8.6/2.0/4.4) and trace elements, but not containing sulfur. Plants were used for experiments 50 d after sowing (DAS), thus 36 d after transfer to experimental soil conditions. We used leaves of rank 5 to 7, which are the mature source leaves with maximum photosynthetic capacity at this developmental stage.

**Gas exchange and sampling.** Plants used for gas-exchange and labelling were taken from the glasshouse at fixed time (4 h after the onset of light) so as to avoid potential diel cycle variation. Gas-exchange under controlled $O_2$/$CO_2$ conditions was performed in a chamber coupled to the LI-COR 6400-XT (LI-COR Biosciences, USA) and having soft walls allowing instant sampling by liquid nitrogen spraying, as described previously[36]. The leaf chamber was adapted to individual leaves with a surface area of about 100 $cm^2$. Light was supplied by an LED panel RGBW-L084 (Walz, Germany). Gas-exchange conditions were: 400 µmol $m^{-2}$ $s^{-1}$ photosynthetically active radiation (PAR), 80% relative humidity, gas flow 35 L $h^{-1}$, and 21–23 °C air temperature. Isotopic labelling was performed using $^{13}CO_2$ (Sigma-Aldrich, 99% $^{13}$C) and $^{33}$S-suphate (Sigma-Aldrich, 98% $^{33}$S; fed via the petiole) for 2 h after 1 h of photosynthetic induction to reach the photosynthetic steady-state. For all $O_2$/$CO_2$ conditions, two series of experiments were done: with $^{13}CO_2$ + $^{33}SO_4^{2-}$, and with natural $CO_2$ and sulfate. Performing experiments with natural compounds was strictly required for calculations of isotopic enrichment using NMR data. The five $O_2$/$CO_2$ conditions presented here are (%/µmol $mol^{-1}$, ordered by increasing carboxylation-to-oxygenation ratio): 100/380, 21/140, 21/380, 21/800, 2/380 and 0/380. In a separate experiment, leaves were labelled with $^{13}$C-5-methionine (99% $^{13}$C on the methyl group) instead of $^{13}CO_2$ and $^{33}$S-sulfate, in order to examine methionine redistribution, that is, to check whether methionine utilization (to proteins, secondary metabolites, etc.) was small within 2 h in the light. Results are shown in Supplementary Fig. 5. For all $^{13}CO_2$/$^{33}$S-sulfate experiments, seven biological replicates (experiments done twice with $n = 4$ and $n = 3$, here pooled together in graphs) were done for each gaseous $O_2$/$CO_2$ condition. For $^{13}$C-methionine labelling (Supplementary Fig. 5B-C), $n = 3$ biological replicates were done for each gaseous $O_2$/$CO_2$ condition.

**NMR isotopic analyses.** Samples were extracted with perchloric acid in liquid nitrogen. Briefly, the sample was ground with 900 µL perchloric acid 70% and 500 µL maleic acid 0.5 M (i.e. a total of 125 µmol per sample, used as an internal standard). The powder was poured in a 50-mL centrifuge tube and then 10 mL MilliQ water were added. After centrifugation (15,000 × g, 15 min), the pellet was re-extracted with 3 mL perchloric acid 2% and centrifuged. The two supernatants were combined, the pH was adjusted to 5 with potassium hydroxide, a 10 µL aliquot was collected for LC-MS analysis and the sample was frozen-dried. The frozen-dried extract was resuspended in 1.6 mL MilliQ water and centrifuged, and the pH was adjusted to 7. For $^{13}$C-NMR ($^{13}$C-methionine labelling experiment), the sample was mixed with $D_2O$ and analysed. For $^{33}$S-NMR (sulfate labelling experiment), the sample was oxidized first. To do so, 600 µL of the sample was mixed with 650 µL $H_2O_2$ (30% v-v) with 0.5 mg of methyltrioxorhenium VII as the

catalyst. This step oxidized quantitatively methionine to methionine sulfone and cysteine to cysteic acid. Oxidized forms of methionine and cysteine are much more easily visible by $^{33}$S NMR than methionine and cysteine themselves. After 3 h at ambient temperature, the sample was vacuum-dried to remove excess $H_2O_2$ and MTO (which volatilizes under vacuum). The dry extract was then resuspended in 550 μL phosphate buffer (pH 7) and 50 μL $D_2O$ was added. 25 μmol taurine (100 μL at 0.25 M) was added (internal standard for $^{33}$S-NMR). The sample was vortexed, and poured in a 5-mm NMR tube (Z107373, Bruker Biospin). $^{33}$S-NMR analyses were performed at 298 K (25 °C) without tube spinning, using a standard pulse program (zg) with 90° pulses for $^{33}$S (15 μs). Acquisition parameters were: 0.09 s acquisition time, 12.8 k size of FID, and a relaxation delay (D1) of 10 ms. 300,000 scans were done, representing about 12 h analysis per sample. Since the responses of the S atoms at different chemical shifts were not perfectly identical, calibration curves were done with known concentrations of standard metabolites (methionine sulfone, cysteic acid, sulfate). Signals obtained by NMR were deconvoluted as Lorentzian curves to calculate $^{33}$S amounts. $^{13}$C-NMR analyzes were performed at 298 K (25 °C) without tube spinning, using an inverse-gated pulse program (zgig) with 90° pulses for $^{13}$C (10 μs). Acquisition parameters were: 1.3 s acquisition time, 114 k size of FID, and a relaxation delay (D1) of 15 s. 2600 scans were done, representing about 10 h analysis per sample.

**LC-MS isotopic analyses**. Liquid chromatography was performed after Abadie et al.[49]. MS analysis was operated in positive polarity using two different runs: in the full MS scan mode first (for molecular average isotopic enrichments) and then in the AIF mode (All Ion Fragmentation; for positional isotopic enrichment in methionine) with an HCD (Higher energy Collision-induced Dissociation) set at 35% (mass scan range 50–750 m/z). The following source settings were used for both scan modes: source voltage 3500 V, resolution 70,000, AGC target 1·10⁶, mass scan range 60–600 m/z, sheath gas 40, auxiliary gas 10, sweep gas 1.5, probe temperature 300 °C, capillary temperature 250 °C and S-lens RF level 50. Equations associated with the calculation of isotopic enrichment are provided in Supplementary Notes 1.

**Metabolomics**. GC-MS analyses were carried out as in ref. [50], using gas chromatography coupled to mass spectrometry (GC-MS), via methanol:water extraction followed by derivatization with methoxyamine and *N*-methyl-*N*-(trimethylsilyl) trifluoroacetamide (MSTFA) in pyridine. Data were extracted using the online processing software Metabolome Express.

**Flux calculations**. $^{33}$S, $^{13}$C isotopic data (absolute contents in mmol m⁻² and %) were used to compute the flux of leaf sulfate reduction, in μmol m⁻² s⁻¹. Equations used and model description are provided in Supplementary Notes 2 and 3.

**Reporting summary**. Further information on research design is available in the Nature Research Reporting Summary linked to this article.

## Data availability

All data generated or analysed during this study are included in this published article in figures (and its supplementary information files), or are available from the corresponding author on reasonable request. Metabolomics data are available on the public Metabolome Express server (www.metabolome-express.org), with the accession reference 33SCOMPLET. The source data underlying Figs. 2 and 3 are shown in Supplementary Data 1.

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

## Acknowledgements

The authors thank the Joint Mass Spectrometry Facility and the NMR Facility of the Australian National University for giving access to instruments. The authors also thank Shoko Okada (CSIRO) for GC-MS analyses of standard compounds. The authors thank the financial support of the Australian Research Council via a Future Fellowship, under contract FT140100645.

## Author contributions

C.A. conducted experiments and data acquisition; G.T. carried out data integration and modelling; C.A. and G.T wrote the paper.

## Competing interests

The authors declare no competing interests.
