## [Peer Review File · Communications Biology]

Editorial Note: This manuscript has been previously reviewed at another journal. This document only contains reviewer comments and rebuttal letters for versions considered at *Communications Biology*.

REVIEWERS' COMMENTS:

Reviewer #2 (Remarks to the Author):

This study examines the relationship between carbon fixation and leaf sulphur assimilation in sunflower plants exposed to atmospheres containing various relative amounts of CO₂ and O₂. The authors conclude that conditions that inhibit photorespiration--i.e., high CO₂ or low O₂ atmospheres--also inhibit shoot sulphur assimilation. They do not, however, provide a thorough explanation for the interdependence of these two processes.

The authors may wish to consider in more detail the parallels between shoot sulphur and nitrate assimilation. Both pathways depend on photorespiration, whereby conditions that inhibit photorespiration, inhibit sulphur and nitrate assimilation in shoots, but stimulate assimilation in roots (Gerlich et al. 2018. *Plant Physiol* 178:565-582; Kruse et al. 2002. *J Exp Bot* 53:2351-2367). These pathways are among the most energetically intensive in plants, and photorespiration generates reductant that may empower both pathways (Scheibe R 2004. *Physiol Plant* 120: 21-26). Carbon fixation may more competitive for reductant than shoot sulphur and nitrate assimilation.

Reviewer #3 (Remarks to the Author):

I have reviewed this manuscript before I find the work described to be technically excellent, the approach taken correct and the significance of the findings correct for this journal.

I do feel that the literature concerning photorespiration and S needs to be better engaged though.

Reviewer #2

This study examines the relationship between carbon fixation and leaf sulphur assimilation in sunflower plants exposed to atmospheres containing various relative amounts of CO₂ and O₂. The authors conclude that conditions that inhibit photorespiration--i.e., high CO₂ or low O₂ atmospheres--also inhibit shoot sulphur assimilation. They do not, however, provide a thorough explanation for the interdependence of these two processes.

This comment has also been articulated by referee 3, who suggested to add more on the literature concerning photorespiration and S. This has now been fixed, with an extended section on this aspect, including the references suggested below. We also have moved the supplementary notes to main text, and they address these aspects (metabolic regulation, link with C₄ photosynthesis).

The authors may wish to consider in more detail the parallels between shoot sulphur and nitrate assimilation. Both pathways depend on photorespiration, whereby conditions that inhibit photorespiration, inhibit sulphur and nitrate assimilation in shoots, but stimulate assimilation in roots (Gerlich et al. 2018. *Plant Physiol* 178:565-582; Kruse et al. 2002. *J Exp Bot* 53:2351-2367). These pathways are among the most energetically intensive in plants, and photorespiration generates reductant that may empower both pathways (Scheibe R 2004. *Physiol Plant* 120: 21–26). Carbon fixation may more competitive for reductant than shoot sulphur and nitrate assimilation.

Thank you for giving us references that have now been included in the paper. We have also re-emphasized the parallelism between sulphate and nitrate assimilation, the latter being proved to decrease as photorespiration decreased.

Reviewer #3

I have reviewed this manuscript before I find the work described to be technically excellent, the approach taken correct and the significance of the findings correct for this journal. Thank you for mentioning the technical excellence of our work.

I do feel that the literature concerning photorespiration and S needs to be better engaged though. This comment has also been articulated by referee 2, who suggested to add more on the literature concerning the interdependence between photorespiration and S. This has now been fixed, with an extended section on this aspect. We also have moved the supplementary notes to main text, and they address these aspects (metabolic regulation, link with C₄ photosynthesis).